# Drawing Reliable Conclusions with Synthetic Simulations from Large Language Models

## Abstract

There is increasing interest in using large language models to generate synthetic simulations (e.g., social simulations) to support social science and human subject research, such as in responses to surveys or in human behavior simulation. However, it is not immediately clear by what means practitioners can incorporate such data alongside ground-truth human data and yet still draw reliable insights and conclusions upon them. In this work, we introduce a principled framework for reliably incorporating synthetic simulations from text-based foundation models into downstream statistical analyses. Our estimator offers a hyperparameter-free solution with strong theoretical guarantees, allowing practitioners to retain key statistical properties—even when incorporating imperfect, biased simulated data. We empirically validate the finite-sample performance of our estimator, which improves statistical efficiency, across different regression tasks in social science applications. To the best of our knowledge, our framework provides the first theoretically-sound approach for safely incorporating synthetic simulations from foundation models for reliable statistical inference.

## 1 Introduction

Recently, practitioners have started to explore the possibility of leveraging large language models to generate synthetic simulated samples, often referred to as social simulations e.g., simulating human responses to surveys or human participants in early pilot studies (Argyle et al., 2023; Brand et al., 2023; Dominguez-Olmedo et al., 2024; Anthis et al., 2025; Hwang et al., 2025b). This has opened up new opportunities to understand human collective behavior, while overcoming the practical limitations and cost restrictions of relying solely on human participants (Alemayehu et al., 2018), leading to recent discourse on how foundation models will transform social science. While most of these studies focus on qualitative takeaways and early signals for future experiments, we focus on the forward-looking setting of making statistically valid inference given such synthetic simulations.

In settings where synthetic simulations are combined with limited human subject data, it is essential that their inclusion does not compromise the validity or reliability of resulting conclusions. To ensure the responsible and safe use of synthetic simulations in such pipelines, we would like to realize the benefits of incorporating information from these additional data sources, while retaining good statistical properties—consistency and proper asymptotic coverage— that are necessary for practitioners to report conclusions reliably. A persistent challenge, however, is that naively combining such imperfect surrogates with ground-truth human samples often introduces substantial bias, leading to significantly biased estimates and compromising the reliability of the conclusions drawn from them.

Yet, existing methods have not made clear by what means we can incorporate such data, while producing statistically valid estimates. First of all, it is not immediately obvious how to even produce synthetic simulations from LLMs such that they can be used in a principled manner. Naively drawing samples from a generative model and treating them as additional samples alongside real data makes it impossible to provide statistical guarantees for the resulting estimate if the generative model does not perfectly match the real distribution— which is expected in practice.

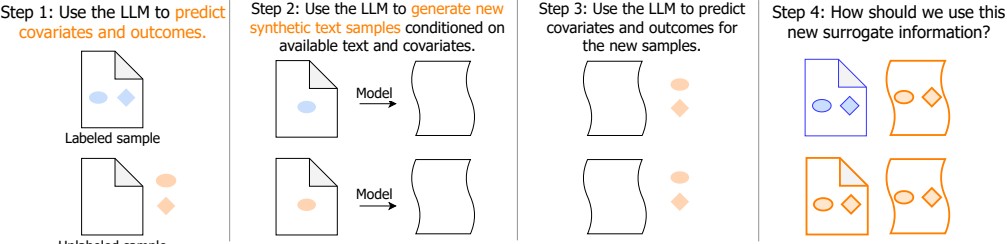

Figure 1: An illustration of our setting. We have real text samples (folded paper icons), some of which are labeled (i.e., have human-annotated covariates and outcomes (blue oval and diamond)). The LLM is used (1) to predict covariates and outcomes (orange oval and diamond), and (2) to generate new, fully synthetic simulations (wavy paper icons). In this work, we refer to social simulations as synthetic simulated samples and synthetic samples, interchangebly.

To address this, we propose a sampling strategy that enables practitioners to leverage synthetic simulations from LLMs in a reliable, principled manner. Concretely, each synthetic sample is generated conditional on an individual real text as an example (see Step 2 illustrated in Figure 1; Section 3 for details), following the common practice of in-context learning (Brown et al., 2020). What makes this formulation statistically powerful is that it introduces a correlation structure between each real text and synthetic sample. Crucially, this correlation structure will prove critical for principled methods for integrating synthetic data, as it enables us to more effectively share information across them (see Section 4.5).

We consider the following setting. We assume the practitioner has access to a corpus of unlabeled text and only a small set of human annotations of covariates and outcomes. In this low-label regime, we examine two increasingly common ways LLMs are used to augment limited-data pipelines—namely, for annotation and generation. More concretely, the practitioner can leverage LLMs to (1) predict covariates and outcomes for the unlabeled text samples; and (2) generate new text samples (i.e., synthetic simulations) conditioned on available samples and extract covariates and outcomes from them similarly to (1) (see Figure 1). Throughout this work, we refer to such LLM-generated samples as synthetic simulations, social simulations, or synthetic data interchangeably. Under this setting, our goal is to study how to effectively combine these different sources of information (from humans and from LLMs) for downstream statistical estimation tasks.

Our primary methodological contribution is to propose a statistically valid estimator that reliably incorporates synthetic simulated samples from LLMs for downstream inference tasks. To the best of our knowledge, this is the first estimator with formal guarantees that solves the problem at hand. The construction of our estimator is based on generalized method of moments (GMM), where we define separate parameters and moments for each data source. While it is not initially obvious that the incorporation of moments based exclusively on synthetic data should yield any benefits (*or even affect*) the estimation of the target parameter, we strikingly find that the interactions between the error residuals of the different sources of information greatly improves estimation (see Sections 4.5 and 5). In other words, our estimator enables practitioners to realize the benefits of LLM-generated synthetic simulated samples, while preserving key statistical properties, necessary for drawing reliable conclusions upon them.

**Contributions.** We (1) introduce a new estimator that incorporates fully synthetic simulated samples from LLMs for downstream inference tasks; (2) provide strong theoretical guarantees on consistency and valid asymptotic coverage; (3) empirically validate its finite sample performance across different regression tasks in LLMs for social science applications; and (4) offer a theoretical analysis to explain how our GMM-based solution obtains these benefits. To the best of our knowledge, our framework provides the first principled approach for incorporating synthetic simulations from large language models for reliable, downstream statistical analyses.

## 2 Related Work

**LLMs for Data Annotation and Synthetic Simulation Tasks.** Our work is motivated by the increasingly growing use and future promise of foundation models (i.e., LLMs) for annotations and simulation studies, particularly as a means to reduce human labeling costs (Hwang et al., 2025a). Recently, LLMs have been tested in fully synthetic simulation studies (Dillion et al., 2023; Anthis et al., 2025), with primary applications in exploratory research or early pilot studies. For instance, recent work has studied simulating individuals in society and their interactions (Park et al., 2022; Chen et al.), analyzing whether the resulting LLM agents produced accurate responses on surveys and accurately predicted behavioral outcomes (Park et al., 2023). Other works have applied LLMs to simulate survey responses (Geng et al., 2024; Rothschild et al., 2024), while others have cautioned about specific flaws in LLM responses (Dominguez-Olmedo et al., 2024), such as not accurately reflecting the influence of demographic groups (Dominguez-Olmedo et al., 2024; Wang et al., 2025). In summary, this line of work shows the potential of synthetic experiments powered through strong generative models but also exhibits clear failure modes and imperfect conclusions from such studies. While most of these studies focus on qualitative takeaways and early signals for future experiments, we focus on the challenging and forward-looking setting of making statistically valid inference given such synthetic simulations.

**Statistical Inference and Debiasing Methods.** Our work is broadly related to performing statistical inference with missing data, where past works have explored approaches to yielding valid and efficient parameter estimates (Robins et al., 1994). Other work has notably explored the usage of ML models to estimate nuisance parameters (Chernozhukov et al., 2018). The most related line of research are debiasing methods (Egami et al., 2023; Gligorić et al., 2024) that focus on combining ground truth data with surrogate predictions (often produced by a machine learning model) to perform statistical inference. These frameworks are often referred to as prediction-powered inference (Angelopoulos et al., 2023a;b) in the machine learning literature. A key difference between these works and our setting is that the primary focus of our work is how to incorporate fully synthetic samples, which remains unaddressed by previous work.

## 3 Preliminaries

**Notation and Setup.** We consider a parameter estimation task where the goal is to estimate a target parameter $\theta^\star \in \mathbb{R}^d$. Let $(T, X, Y) \sim \mathcal{D}$ denote a random triple drawn from an unknown data-generating distribution $\mathcal{D}$ over text inputs $T \in \mathcal{T}$, covariates about the text (e.g., structured metadata) $X \in \mathcal{X} \subseteq \mathbb{R}^d$, and labels $Y \in \mathcal{Y}$. For example, $T$ can be texts from online requests, where $X$ are linguistic markers of hedging (i.e., notions of uncertainty) and $Y$ is perceived politeness. Due to labeling budget constraints, we assume we only observe a small fraction of human-annotated data (i.e., ground-truth covariates and labels about the text). Specifically, we have access to labeled dataset $\mathcal{D}_{\text{labeled}} = \{(T_i, X_i, Y_i)\}_{i=1}^n$ that is sampled i.i.d. from $\mathcal{D}$ and an unlabeled corpus of text $\mathcal{D}_{\text{unlabeled}} = \{(T_j)\}_{j=n+1}^{n+m}$ sampled i.i.d. from $\mathcal{D}_T$ (i.e., the marginal distribution over $T$), where $m \gg n$. To supplement this limited supervision, we leverage machine learning models (i.e., text-based foundation models) in the following two ways.

**Proxy Covariates and Labels.** We use a machine learning model $f$ to produce predictions $\{f_X(T_j), f_Y(T_j)\}$ for the available set of input texts $T \in \mathcal{T}$. Here, $f_X$ and $f_Y$ denote the same machine learning model, using separate prompts for the target outcome (either a covariate $X$ or outcome $Y$) (see Appendix E for details). This yields the following $\mathcal{D}_{\text{proxy}} = \{(T_i, f_X(T_i), f_Y(T_i))\}_{i=1}^n \cup \{(T_j, f_X(T_j), f_Y(T_j))\}_{j=n+1}^{n+m}$. For simplicity, we will refer to this as **proxy samples** and denote them as $(T, \hat{X}, \hat{Y})$. We will refer to the distribution over proxy samples as $\hat{\mathcal{D}}$. Note that this is the setting previous works have considered (mainly restricted to predicted outcomes) when addressing this problem.

**Synthetic Covariates and Labels.** We propose a new data augmentation process which generates new samples using a text-based foundation model (employing it as a generative model, instead of a classifier as in previous works studying the proxy setup). Specifically, our method conditions the generation process on each individual text $T_j$ as an example and asks the model to generate a new synthetic sample given that context. Formally, for each $i$, we sample a new text $\tilde{T}_i$, conditioned on $(T_i, X_i)$ if the sample is labeled and $(T_j, \hat{X}_j)$ if the sample is unlabeled. For example, "Consider text taken from user requests on Stack Exchange, either containing a hedging device or not containing one. {Insert example $T_i$ and covariate $X_i$}. Now, generate a new example of a request that matches the style of the provided example."[1] Based on the generated sample, which we denote as $\tilde{T}_i$, we then extract its corresponding covariates and outcomes similarly as in proxy samples. More concretely,

$$\tilde{T}_k \sim \mathbb{P}(\cdot \mid T_i, X_i) \text{ if labeled,} \qquad \tilde{X}_k \sim \mathbb{P}(\cdot \mid \tilde{T}_k),$$
$$\tilde{T}_k \sim \mathbb{P}(\cdot \mid T_j, \hat{X}_j) \text{ if unlabeled} \qquad \tilde{Y}_k \sim \mathbb{P}(\cdot \mid \tilde{T}_k)$$

resulting in the following $\mathcal{D}_{\text{synthetic}} = \{(\tilde{T}_k, \tilde{X}_k, \tilde{Y}_k)\}_{k=1}^{n+m}$. We will refer to the distribution over **synthetic samples** $(\tilde{T}, \tilde{X}, \tilde{Y})$ as $\tilde{D}$.

This specific sampling process has two motivations. First, from a machine learning perspective it can be seen as a form of in-context prompting, where the model is given an example from the dataset in order to align it more closely with the task. Iteratively prompting with different samples $T_i$ is also likely to produce more diverse samples than asking for many samples with the same prompt. Second, from a statistical perspective, it introduces a correlation structure between each real text $T_i$ and synthetic sample $\tilde{T}_i$. This correlation structure will prove critical for principled methods for integrating synthetic data because it allows us to more effectively share information across them. Indeed, naively drawing a set of synthetic samples from the generative model and pooling them with the real data would render it impossible to provide statistical guarantees for the resulting estimate if generative model fails to perfectly match the real distribution.

Finally, we introduce some notation that combines all of these data sources into draws from a single joint distribution. Specifically, we introduce a new random variable $s \in \{0, 1\}$ which is an indicator for whether $T$ is labeled (1) or unlabeled (0). Then, we view the complete generative process as draws $(T, s, s \cdot X, s \cdot Y, \tilde{X}^1, \tilde{Y}^1 ... \tilde{X}^M, \tilde{Y}^M)$ for $M$ different kinds of auxiliary data. So far, we have discussed two kinds, proxy and synthetic, that we employ empirically ($M = 2$), but our methods are fully extensible to additional kinds of auxiliary data. For example, we could include samples from multiple different generative models. The real $(X, Y)$ are observed only for labeled points with $s = 1$ while the auxiliary data is available for all samples. The joint distribution over this full tuple is induced by the composition of the generative processes for the components described above.

# 4 Combining Synthetic Information via Generalized Method of Moments

To estimate the target parameter $\theta^\star$, we propose an approach based on generalized method of moments (GMM) (Hansen, 1982) that combines information from the different types of data in the following manner.

## 4.1 Moment Conditions

Our framework is applicable whenever the target parameter can be identified by a set of moment conditions, functions whose expectation should be zero at the true value of the parameter. Moment-based estimation is a broad and flexible framework that includes almost all commonly used statistical frameworks (e.g., maximum likelihood, generalized linear models, instrumental variables, etc). We begin by defining the moment conditions

---

[1]See Appendix E for further prompt details.

that identify $\theta^*$ under the distribution of interest (i.e., the real-data distribution $\mathcal{D}$). In the following section, we introduce how this can be adapted to incorporate surrogate data (i.e., proxy and synthetic data).

Formally, we consider the problem of estimating a parameter $\theta \in \mathbb{R}^d$. The true value $\theta^*$ is identified as the solution to a set of $p \geq d$ moment conditions

$$\mathbb{E}[\psi^{(\ell)}(\theta^*)] = 0, \quad \ell = 1...p$$

where the $\psi^{(\ell)}$ are continuously differentiable functions $\mathbb{R}^d \to \mathbb{R}$. For example, in a maximum likelihood model, we would have one $\psi$ for the derivative of the log-likelihood with respect to each parameter, and the moment conditions enforce that $\theta^*$ satisfies the first-order conditions for maximizing the likelihood. Let $\psi(\theta) = [\psi^{(1)}(\theta)...\psi^{(p)}(\theta)]^\top$ denote a column vector stacking the $p$ moments.

## 4.2 Constructing Our GMM Estimator

To leverage the auxiliary data (i.e., proxy data and synthetic data) in making our GMM estimator more efficient, we can construct a set of auxiliary moments for each additional source of data. We estimate an additional set of auxiliary parameters $\eta_1, ..., \eta_M \in \mathbb{R}^p$, one parameter vector for each set of new auxiliary data. In the specific instantiation of the model that we use here, we always have $M = 2$ (proxy and synthetic data), but in principle our method is extensible to many sources of auxiliary data, for example synthetic samples generated from several different models. Roughly, each new parameter vector $\eta_i$ can be understood as the parameter that we would estimate using each auxiliary data source, and our augmented model will automatically determine how to use these auxiliary estimates to inform the estimate of the parameter of interest $\theta$.

For each new parameter vector $\eta_i$, we introduce a corresponding set of new moments to estimate this parameter and allow its estimate to inform the estimate of $\theta$. Specifically, we introduce for each $\eta_i$ two new blocks of moments that are copies of the original moments for $\theta$. Intuitively, one block of moments will be evaluated only on the real (labeled) data, while the other will be taken on the pooled set of labeled data and auxiliary dataset $i$. The pooled-data moment will allow us to improve the estimation of $\eta_i$ using the larger sample. The version evaluated only on the real data will allow GMM to evaluate how well the moments for the auxiliary parameter correlate with those of the true parameter on the same data, and share information across them if the auxiliary moments are informative (as we would expect if the generated data is high quality).

Formally, let $S_t \in \mathbb{R}^p$ stack $p$ copies of the indicator variable $s_t$ for whether a data point $t$ is labeled. In block matrix notation, the combined model takes the form of the augmented moments

$$g_t(\theta, \eta) = \begin{bmatrix} S_t \\ S_t \\ \vdots \\ S_t \\ 1 \\ \vdots \\ 1 \end{bmatrix} \odot \begin{bmatrix} \psi(\theta) \\ \psi(\eta_1) \\ \vdots \\ \psi(\eta_M) \\ \psi(\eta_1) \\ \vdots \\ \psi(\eta_M) \end{bmatrix} \in \mathbb{R}^{p+2Mp} \tag{1}$$

We will then jointly estimate $(\theta, \eta)$ as the solution to the moment condition $\mathbb{E}[g_t(\theta, \eta)] = 0$. For clarity, we refer to our estimator that uses real and proxy data ($M = 1$) as **GMM-Proxy** and our estimator that uses real, proxy, and synthetic data ($M = 2$) as **GMM-Synth** throughout the paper. See Appendix B for further details. We remark that since the parameter of interest $\theta$ appears only in its original set of moments, which are evaluated only on the labeled data, this new moment condition still identifies the target parameter $\theta^*$. However, as we discuss below, when we apply standard methods for efficiently estimating

the augmented GMM, the new moment conditions will be leveraged to reduce the variance of the estimate without compromising consistency or asymptotic normality.

## 4.3 GMM Estimation

Given our augmented moment conditions $g$, we estimate the parameters $(\theta, \eta)$ by minimizing the GMM objective:

$$\hat{\theta}_T, \hat{\eta}_T = \arg \min_{\theta \in \Theta, \eta \in \mathbb{R}^{2Mp}} \widehat{Q}_T(\theta, \eta), \tag{2}$$

where

$$\widehat{Q}_T(\theta, \eta) = \left[ \frac{1}{T} \sum_{t=1}^{T} g_t(\theta, \eta) \right]^{\top} \widehat{\mathbf{W}}_T \left[ \frac{1}{T} \sum_{t=1}^{T} g_t(\theta, \eta) \right]. \tag{3}$$

Here, $\widehat{\mathbf{W}}_T \in \mathbb{R}^{M \times M}$ is a (possibly data-dependent) positive semidefinite weighting matrix that determines the importance of each moment condition in the estimation objective. While GMM estimators are consistent and normal under *any* choice of positive definite $\widehat{\mathbf{W}}_T$, the selection of $\widehat{\mathbf{W}}_T$ influences their efficiency.

**Two-step GMM estimator.** We adopt the two-step GMM procedure as described in Newey & McFadden (1994). First, we compute the one-step estimator $\hat{\theta}_T^{(\text{os})}, \hat{\eta}_T^{(\text{os})}$ using an identity weight matrix $\widehat{\mathbf{W}}_T = \mathbf{I}$. Then, we estimate the optimal weight matrix as:

$$\widehat{\Omega}_T(\hat{\theta}_T^{(\text{os})}, \hat{\eta}_T^{(\text{os})}) = \left[ \frac{1}{T} \sum_{t=1}^{T} g_t(\hat{\theta}_T^{(\text{os})}, \hat{\eta}_T^{(\text{os})}) g_t(\hat{\theta}_T^{(\text{os})}, \hat{\eta}_T^{(\text{os})})^{\top} \right], \tag{4}$$

and set

$$\widehat{\mathbf{W}}_T = \left[ \widehat{\Omega}_T(\hat{\theta}_T^{(\text{os})}, \hat{\eta}_T^{(\text{os})}) \right]^{-1}. \tag{5}$$

This optimal weighting has the interpretation as the inverse empirical covariance of the moment conditions on the one-step estimate. We then compute the final two-step estimator by minimizing $\widehat{Q}_T(\theta)$ with this updated weighting matrix. This choice of $\widehat{\mathbf{W}}_T$ yields an asymptotically efficient estimator under standard GMM regularity conditions.

The adoption of two-step GMM is a critical component of our proposed estimation framework. Indeed, in the first-step estimates, the synthetic and proxy data will have no impact on the estimate of $\theta$ because they never appear in the moment conditions concerning $\theta$. In the second stage though, the weight matrix $\widehat{\mathbf{W}}_T$ accounts for the covariance between moment conditions, where off-diagonal terms in the matrix allow moments for the auxiliary data sources to influence the estimation of $\theta$.

## 4.4 Consistency and Asymptotic Inference

We now present results on the consistency and asymptotic behavior of our GMM estimators.

**Proposition 1.** *Our estimate $\hat{\theta}_T$ (as defined in Equation 3) is consistent and asymptotically normal. It converges in distribution as*

$$\sqrt{T}((\hat{\theta}_T', \hat{\eta}_T')' - (\theta', \eta')') \xrightarrow{d} \mathcal{N}(0, V)$$

*where the covariance $V$ is given by*

$$V = \left( G(\theta, \eta)^T \widehat{\mathbf{W}} G(\theta, \eta) \right)^{-1}$$
$$G(\theta, \eta)^T \widehat{\mathbf{W}} F \widehat{\mathbf{W}} G(\theta, \eta)$$
$$\left( G(\theta, \eta)^T \widehat{\mathbf{W}} G(\theta, \eta) \right)^{-1},$$

*and where $G(\theta, \eta)$ is the Jacobian of the population moments at the ground truth parameter values $\theta, \eta$.*

For optimal weight matrix in Equation 5, this simplifies to $V = (G(\theta, \eta)^T F^{-1} G(\theta, \eta))^{-1}$. These are standard results on GMM estimators, which follow by straightforwardly applying the results in Hansen (1982). We remark that these asymptotic results require a set of conditions on the sample moments, which are slightly nuanced in this setting with multiple sources of information. We discuss these conditions and prove that they are satisfied in Appendix A for the setting of proxy and synthetic samples. Given this asymptotic behavior, we can derive valid confidence intervals for our parameter estimates.

### 4.5 Why does synthetic data improve performance?

To understand where the benefits arise from incorporating the proxy and synthetic data into our GMM estimator, we analyze the interaction between our moment conditions. Note that the functions $\psi$ are often referred to as "residuals" in the GMM literature; since $\psi(\theta)$ should be zero in-expectation, deviations from zero are interpretable as a kind of residual. The key intuition is that synthetic data will improve performance when the synthetic-data residuals are predictive of the real-data residuals.

First, we note that if the synthetic data were perfectly simulated, $X$ and $Y$ would be perfectly recovered from the unlabeled text $T$. With ground truth $X, Y$, we can perfectly recover the residual terms. In settings where we have good but imperfect simulations, $\hat{X}, \hat{Y}$ and $\tilde{X}, \tilde{Y}$ are highly correlated with the errors in the true data, and we can approximately estimate the real-data residuals with the synthetic data. Within our GMM-based approach, this is all handled implicitly in our two-step estimation procedure. During the first estimation step, each set of parameters (e.g., defined on the observed, proxy, and synthetic data) is independently identified since the initial weighting is an identity matrix. The key insight is that, during the second estimation step, the weighting matrix $\widehat{W}$, which is the inverse of the moment covariance matrix, captures the interactions between the observed residual terms and the residuals from the synthetic data in our GMM objective.

Partitioning the moments into observed data residuals $m_t(\theta)$ and synthetic data residuals $h_t(\eta)$, we derive an explicit formula for the asymptotic variance of $\sqrt{T}(\hat{\theta}_T - \theta)$ in Appendix C. We find two important conclusions. First, when these residuals are independent of the observed data, the formula reduces to the optimal variance based only on the fully observed data. That is, in the worst case where synthetic data is completely uninformative, including it does not hurt (at least asymptotically). Second, when the real and synthetic residuals are correlated (as we would hope), we derive a lower bound on the variance which is proportional to the residual variance in a regression of the observed data residuals on the span of the synthetic data residuals. This bound is minimized by choosing moments that span the conditional expectation of the observed data residuals given $T_i$, a sufficient condition for which is that the conditional distribution of $\hat{X}, \hat{Y}$ or $\tilde{X}, \tilde{Y}$ given $T$ equals the conditional distribution of $X, Y$.

## 5 Experimental Results

We evaluate the finite-sample performance of our proposed estimators (GMM-Synth and GMM-Proxy) as well as the adapted debiasing-based estimators (PPI++Synth and PPI++Proxy) (see Appendix D) in the following setup.

**Datasets and Experimental Setup.** We focus on the small-data regime, where the need for additional data sources is especially well-motivated. In particular, we consider settings where the practitioner has a corpus of unlabeled text and only a small set of human-annotated samples (e.g., ground-truth covariates and labels derived from the text). We evaluate our framework in four different computational social science tasks, each involving a regression coefficient as the target quantity. In the first two tasks, we use texts from online requests posted on Stack Exchange and Wikipedia (Danescu-Niculescu-Mizil et al., 2013) to estimate how certain linguistic features affect perceived politeness; specifically, the use of first-person plural pronouns and the presence of hedging markers (i.e., expressions of uncertainty). The third task examines the effect of affirming linguistic devices on media

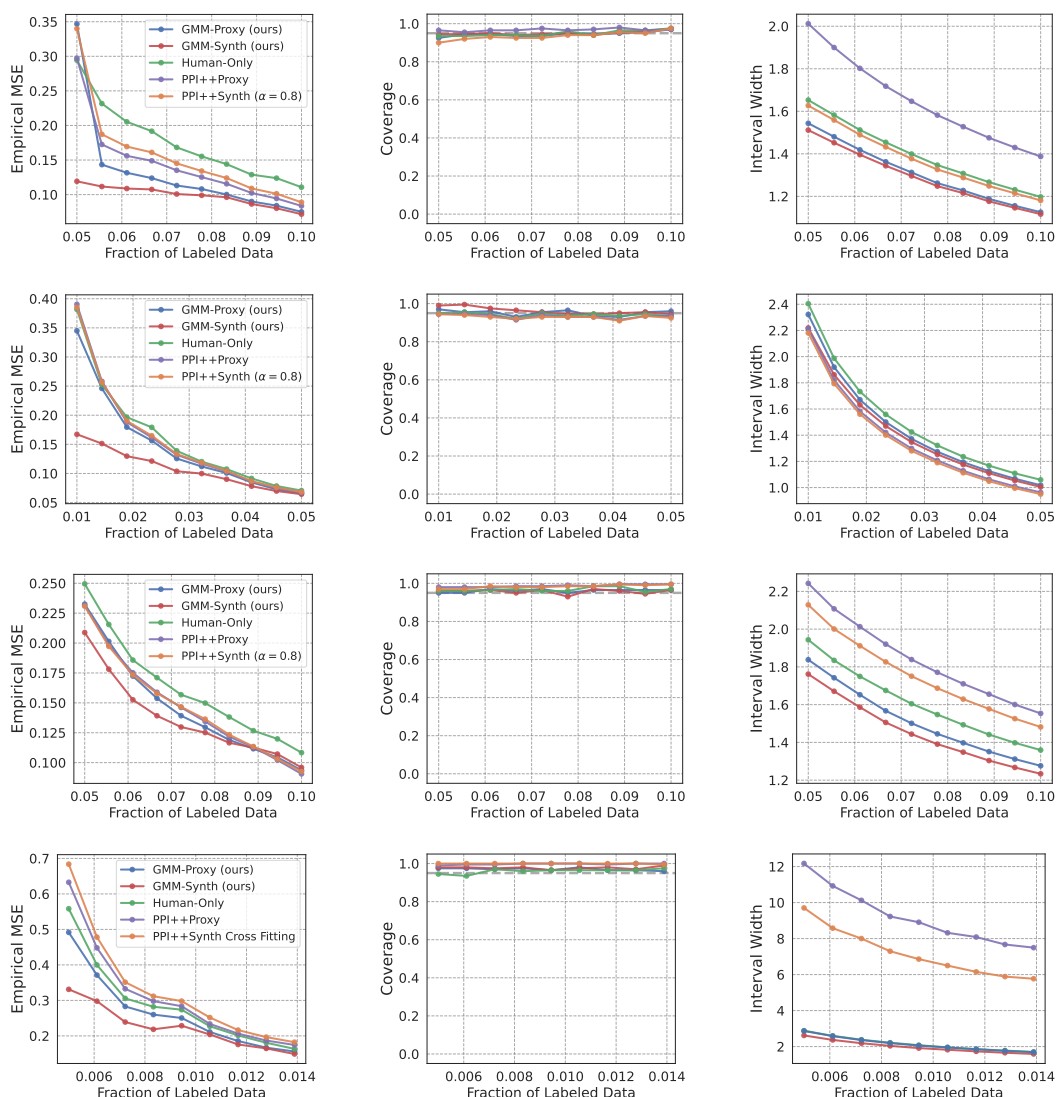

Figure 2: **Main Results**. We observe large reductions in MSE, especially in very low-label regimes. Each row corresponds to a task (i.e., 1pp, Hedging, Stance, Congressional Bills (from top to bottom)); each column corresponds to a metric (i.e., MSE, coverage, confidence interval width (from left to right)). Note that when the best performing PPI++Synth is equivalent to PPI++Proxy, we report the second-best performing PPI++Synth method ($\alpha = 0.8$ for these tasks). Results are averaged over 200 trials.

stance toward global warming (i.e., whether the news headline supports or rejects climate change) using a corpus of climate-related news headlines (Hmielowski et al., 2014). Finally, in the fourth task, we analyze congressional bills texts (Adler & Wilkerson, 2011) to estimate the effect of a legislator's DW-Nominate measure (Lewis et al., 2024) of ideology on the type of bill (whether the bill pertains to macroeconomy). In all the tasks, the target quantity is the regression coefficient corresponding to the explanatory variable of interest.

To evaluate our framework, we use GPT-4o (Hurst et al., 2024) to generate proxy and synthetic data, without any task-specific fine-tuning, i.e., using the LLM out of the box. We report the empirical mean-squared error (MSE), coverage at level $\alpha = 0.05$, confidence interval width, and effective sample size across all tasks. The effective sample size $n_{\text{effective}}$ denotes the number of human-labeled samples needed for the classical estimator $\hat{\theta}^{\text{human}}$ to match the MSE of the method's estimate $\hat{\theta}^{\text{method}}$. In other words, it quantifies how many

human annotations the method effectively saves while maintaining equivalent accuracy. We defer the results and discussion for effective sample size results to Appendix F (see Figure 4).

**Key Observations.** We begin by presenting our main empirical results. In Figure 2, we evaluate the performance of our GMM-based estimators: GMM-Proxy and GMM-Synth. Across all studied tasks, we observe both methods consistently outperform only leveraging ground-truth human-annotated samples (Human-only), yielding improvements in both point estimation (MSE) and inference (tighter intervals while retaining proper coverage). We observe that these gains are especially pronounced in low-label regimes, which precisely aligns with the motivating use case of our framework. On several tasks (e.g., 1pp, Hedging, and Congressional Bills), in low-label regimes, we observe large reductions in MSE, often exceeding 50% reductions compared to the human-only baseline. Furthermore, in Figure 4 (see Appendix F), we observe that our GMM-based approaches consistently improve performance in terms of effective sample size across all tasks. That is, our method reduces the number of human annotations needed to achieve *equally accurate* estimates. This is particularly valuable in label-scarce settings, highlighting its practical value for practitioners in low-resource, limited-labeled regimes.

Interestingly, these gains cannot be explained by the proxy or synthetic data alone as both sources produce greatly biased estimates (see Figure 5; Appendix F). This again highlights the detrimental risks of naively using LLM-simulated data in such pipelines. The key to attaining these benefits lies in the specific structure of *how we combine these data sources* with human-labeled data. The key intuition is that synthetic data will improve performance when the synthetic-data residuals are predictive of the real-data residuals. See Section 4.5 for a deeper analysis of how this interaction improves performance.

We next examine the performance of our adapted debiasing-based estimators: PPI++Proxy and PPI++Synth. Note that in the implementation of our debiasing-based estimators, we leverage PPI++ (Angelopoulos et al., 2023b), which further includes benefits of power tuning. Empirically, we find that PPI++Synth often underperforms, due to cross-fitting restricting the sample size even further (see Figure 6; Appendix G for details). In Figure 2, we observe that although both methods retain reasonable coverage, they systematically underperform the GMM-based estimators, producing larger MSE and mostly wider intervals. Most notably, our findings show that while debiasing-based approaches effectively incorporate proxy data, they struggle to leverage fully synthetic data, yielding negligible to no improvement when such data is incorporated. Given this limitation, our GMM-based strategy for incorporating synthetic data may be of broader interest as an alternative to the predominant debiasing-based methods used so far in the literature for incorporating biased sources of information.

## 6  Discussion

While pipelines leveraging synthetic simulations have yet to be fully realized, developing reliable mechanisms for integrating these data sources is indeed what will inform *how* such pipelines should be designed and implemented in practice. In this work, we introduce the first principled framework for reliably incorporating synthetic simulated samples into downstream statistical analyses. We provide practical guidance for constructing synthetic simulated samples from text-based foundation models in ways that support valid inference, and propose a new estimator based on generalized method of moments (GMM) estimation, where the key intuition is that synthetic data will improve performance when the synthetic-data residuals are predictive of the real-data residuals. Across the studied regression tasks, we indeed observe a large degree of improvements in estimation, especially in very low-label regimes. More broadly, this work takes a first step toward understanding how imperfect simulated data from foundation models can systematically be leveraged to support valid inference and to make reliable downstream conclusions. As the usage and future promise of foundation models continue to grow, so too will the complexity of pipelines that incorporate their outputs. Our framework provides a foundation for easily extensible estimation methods that can safely incorporate the growing variety and quality of synthetic data sources from such models.

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

## A    Conditions for Consistency and Asymptotic Normality

We provide a discussion about the necessary conditions for a GMM estimator to be consistent and asymptotically normal, showing that these conditions are indeed met for our augmented GMM.

As mentioned in the construction of our estimator, we define one moment condition for each parameter on the observed data $D$. We also define two moments for each parameter on the proxy and synthetic data. This leads to an overidentified system, with more moments than parameters, ensuring that the target parameter is identifiable.

Next, we establish a few conditions for valid asymptotic properties of our GMM estimator, specifically about the convergence and distributions of the sample moments. First, we require that all of our moments converge to their expectation, or that

$$\frac{1}{n} \sum_{i=1}^{n} \psi^{(j)} \to \mathbb{E}[\psi^{(j)}].$$

Next, they must also obey the central limit theorem, or that

$$\sqrt{n} \left( \frac{1}{n} \sum_{i=1}^{n} \psi^{(j)} \right) \xrightarrow{d} \mathcal{N}(0, F),$$

where $F$ is some finite covariance matrix of all the moments.

Under these standard regularity conditions on the moment functions $\psi$ (Newey & McFadden, 1994), these conditions are immediately satisfied for the moments defined on observed data, as each observation of the moments are independent. The same holds for the moments defined on proxy data, since $\hat{X}, \hat{Y}$ are functions of independent inputs $T$, and are therefore also independent across observations. The case of synthetic data is slightly more nuanced, but we show that the required conditions still hold, through the following lemma.

**Lemma 1.** *Let $\{\phi\}_{j=1}^{m}$ represent our moments defined on synthetic observations. Then, they are i.i.d., and consequently*

$$\frac{1}{m} \sum_{j=1}^{m} \phi_j \to \mathbb{E}[\phi_j] \quad and \quad \sqrt{m} \left( \frac{1}{m} \sum_{j=1}^{m} \phi_j \right) \xrightarrow{d} \mathcal{N}(0, \sigma(\phi)),$$

*where $\sigma(\phi)$ is the variance matrix of $\phi$.*

*Proof.* We begin by noting that the unlabeled texts $\{T_j\}_{j=1}^{m}$ are drawn i.i.d. from the marginal distribution $\mathcal{D}_T$. For each $T_j$, a synthetic text $\tilde{T}_j$ is generated by a generative model (i.e., an LLM), which uses independent randomness for each call. The model is conditioned only on an individual sample $(T_j, X_j)$ if $j$ is labeled or $(T_j, \hat{X}_j)$ otherwise. Since the generative process for each $T_j$ is independent and the mapping $\tilde{T}_j \mapsto (\tilde{X}_j, \tilde{Y}_j)$ is applied identically to each sample, the resulting pairs $(\tilde{X}_j, \tilde{Y}_j)$ are also i.i.d. As these pairs are drawn i.i.d., then these conditions are met via the central limit theorem. □

This result shows that the required conditions on the sample moments hold in our setting of proxy and synthetic samples; under the regularity conditions of Newey & McFadden (1994) Theorem 3.2, one immediately obtains Proposition 1 on the asymptotic behavior of our GMM estimator.

## B    Moment Conditions

We provide a concrete example of our moment construction for the case of generalized linear models (GLMs) in two-dimensions.

### B.1 Example 1. Generalized Linear Models

Recall that the standard GLM formulation optimizes the objective function,

$$\ell_\theta(x, y) = -y x^T \theta + f(x^T \theta),$$

where $f$ is a function that is convex and infinitely differentiable. We remark that this recovers the setting of logistic regression when $f(z) = \log(1 + \exp(z))$. Let us assume a two-dimensional setting for illustration. This translates to the population moment conditions of

$$\mathbb{E}\left[X_1\left(Y - \frac{\partial f}{\partial \theta_1}(X^T \theta^*)\right)\right] = 0, \quad \mathbb{E}\left[X_2\left(Y - \frac{\partial f}{\partial \theta_2}(X^T \theta^*)\right)\right] = 0$$

We have similar moments for proxy and synthetic data, where we use parameters $\eta = (\eta^{(1)}, \eta^{(2)})$, which are also two-dimensional. Within our GMM framework, we construct the following set of moment conditions across the observed, proxy, and synthetic data.

$$g_t(\theta, \eta) = \begin{bmatrix} s_t \\ s_t \\ s_t \\ s_t \\ s_t \\ s_t \\ 1 \\ 1 \\ 1 \\ 1 \end{bmatrix} \odot \begin{bmatrix} X_{t,1}(Y_t - \frac{\partial f}{\partial \theta_1}(X_t^T \theta)) \\ X_{t,2}(Y_t - \frac{\partial f}{\partial \theta_2}(X_t^T \theta)) \\ \hat{X}_{t,1}(\hat{Y}_t - \frac{\partial f}{\partial \eta_1^{(1)}}(\hat{X}_t^T \eta^{(1)})) \\ \hat{X}_{t,2}(\hat{Y}_t - \frac{\partial f}{\partial \eta_2^{(1)}}(\hat{X}_t^T \eta^{(1)})) \\ \tilde{X}_{t,1}(\tilde{Y}_t - \frac{\partial f}{\partial \eta_1^{(2)}}(\tilde{X}_t^T \eta^{(2)})) \\ \tilde{X}_{t,2}(\tilde{Y}_t - \frac{\partial f}{\partial \eta_2^{(2)}}(\tilde{X}_t^T \eta^{(2)})) \\ \hat{X}_{t,1}(\hat{Y}_t - \frac{\partial f}{\partial \eta_1^{(1)}}(\hat{X}_t^T \eta^{(1)})) \\ \hat{X}_{t,2}(\hat{Y}_t - \frac{\partial f}{\partial \eta_2^{(1)}}(\hat{X}_t^T \eta^{(1)})) \\ \tilde{X}_{t,1}(\tilde{Y}_t - \frac{\partial f}{\partial \eta_1^{(2)}}(\tilde{X}_t^T \eta^{(2)})) \\ \tilde{X}_{t,2}(\tilde{Y}_t - \frac{\partial f}{\partial \eta_2^{(2)}}(\tilde{X}_t^T \eta^{(2)})) \end{bmatrix}$$

## C Partitioned GMM Asymptotic Variance

We now derive the asymptotic variance of our GMM estimator for specifically the target parameter $\hat{\theta}_T$.

**Theorem 1.** *The asymptotic variance of $\sqrt{T}(\hat{\theta}_T - \theta)$ is given by*

$$\left(\frac{d\mathbb{E}[m(\theta)]}{d\theta'} A \frac{d\mathbb{E}[m(\theta)]}{d\theta} - \frac{d\mathbb{E}[h(\eta)]}{d\eta'} B^\top \frac{d\mathbb{E}[m(\theta)]}{d\theta}\left(\frac{d\mathbb{E}[h(\eta)]}{d\eta'} D \frac{d\mathbb{E}[h(\eta)]}{d\eta}\right)^{-1} \frac{d\mathbb{E}[m(\theta)]}{d\theta'} B \frac{d\mathbb{E}[h(\eta)]}{d\eta}\right)^{-1}.$$

*with $A, B, D$ defined below.*

*Proof.* With the optimal choice of weight matrix for the full GMM estimation problem, the asymptotic variance of the vector $(\hat{\theta}, \hat{\eta})$ converges to $(G^T F^{-1} G)^{-1}$. To obtain the variance for $\hat{\theta}$ specifically, partition the moments into $g_t(\theta, \eta) = (m_t(\theta)', h_t(\eta)')'$, where $m_t(\theta) = S_t \odot \psi(\theta)$, and

$$h_t(\eta) = \begin{bmatrix} S_t \\ S_t \\ \vdots \\ S_t \\ 1 \\ \vdots \\ 1 \end{bmatrix} \odot \begin{bmatrix} \psi(\eta^{(1)}) \\ \vdots \\ \psi(\eta^{(M)}) \\ \psi(\eta^{(1)}) \\ \vdots \\ \psi(\eta^{(M)}) \end{bmatrix}$$

507 Given this partitioning, we can express

$$G(\theta, \eta) = \begin{bmatrix} \frac{d\mathbb{E}[m(\theta)]}{d\theta} & 0 \\ 0 & \frac{d\mathbb{E}[h(\eta)]}{d\eta} \end{bmatrix}$$

$$F = \begin{bmatrix} \mathbb{E}[m_t(\theta)m_t(\theta)'] & \mathbb{E}[m_t(\theta)h_t(\eta)'] \\ \mathbb{E}[h_t(\eta)m_t(\theta)'] & \mathbb{E}[h_t(\theta)h_t(\theta)'] \end{bmatrix}$$

508 By the partitioned inverse formula, we can express $F^{-1}$ as

$$\begin{bmatrix} A & B \\ B^\top & D \end{bmatrix}$$

509 where the upper left block $A$ is

$$(\mathbb{E}[m_t(\theta)m_t(\theta)'] - \mathbb{E}[m_t(\theta)h_t(\eta)']\mathbb{E}[h_t(\theta)h_t(\theta)']^{-1}\mathbb{E}[h_t(\eta)m_t(\theta)'])^{-1}$$

510 This term can be interpreted as the inverse of the asymptotic residual variance of a regression
511 of $m_t(\theta)$ on the span of the vector $h_t(\eta)$.

512 The lower right block $D$ is, symmetrically, the asymptotic residual variance of a regression
513 of $h_t(\theta)$ on the span of the vector $m_t(\eta)$:

$$(\mathbb{E}[h_t(\theta)h_t(\theta)'] - \mathbb{E}[h_t(\theta)m_t(\eta)']\mathbb{E}[m_t(\theta)m_t(\theta)']^{-1}\mathbb{E}[m_t(\eta)h_t(\theta)'])^{-1}$$

514 Finally, the off-diagonal term multiplies $A$ by the coefficient in a regression of $m$ on $h$:

$$B = -A\mathbb{E}[m_t(\theta)h_t(\eta)']\mathbb{E}[h_t(\theta)h_t(\theta)']^{-1}$$

515 For the full variance,

$$G^\top F^{-1} G = \begin{bmatrix} \frac{d\mathbb{E}[m(\theta)]}{d\theta'} A \frac{d\mathbb{E}[m(\theta)]}{d\theta} & \frac{d\mathbb{E}[m(\theta)]}{d\theta'} B \frac{d\mathbb{E}[h(\eta)]}{d\eta} \\ \frac{d\mathbb{E}[h(\eta)]}{d\eta'} B^\top \frac{d\mathbb{E}[m(\theta)]}{d\theta} & \frac{d\mathbb{E}[h(\eta)]}{d\eta'} D \frac{d\mathbb{E}[h(\eta)]}{d\eta} \end{bmatrix}$$

Applying the partitioned inverse formula again, the upper left block of $(G^\top F^{-1} G)^{-1}$, which gives exactly the asymptotic variance of $\sqrt{T}(\hat{\theta}_T - \theta)$, is equal to

$$(\frac{d\mathbb{E}[m(\theta)]}{d\theta'} A \frac{d\mathbb{E}[m(\theta)]}{d\theta} - \frac{d\mathbb{E}[h(\eta)]}{d\eta'} B^\top \frac{d\mathbb{E}[m(\theta)]}{d\theta} (\frac{d\mathbb{E}[h(\eta)]}{d\eta'} D \frac{d\mathbb{E}[h(\eta)]}{d\eta})^{-1} \frac{d\mathbb{E}[m(\theta)]}{d\theta'} B \frac{d\mathbb{E}[h(\eta)]}{d\eta})^{-1}$$

516 This can be interpreted similarly as the asymptotic variance of the residual prediction error
517 from a regression of $A^{-1/2}\frac{dm(\theta)}{d\theta}$ onto the span of a weighted linear combination of terms in
518 $\frac{dh(\eta)}{d\eta}$. $\qquad\square$

We remark that a lower bound on the total variance is given by $(\frac{d\mathbb{E}[m(\theta)]}{d\theta'} A \frac{d\mathbb{E}[m(\theta)]}{d\theta})^{-1}$, which is minimized when $A$ is maximized. Among choices of moment functions $h_t(\eta)$ that depend solely on $T_t$, $A$ is maximized in the positive semi-definite order when the span of $h_t(\eta)$ contains $\mathbb{E}[m(\theta)|T_t]$. A sufficient but not necessary condition for this is that for some $j \in 1 \ldots M$, the conditional moments of the simulation are identical to those of the real data:

$$E[\psi(\eta_j)|T_i] = E[\psi(\theta)|T_i]$$

519 This calibration condition is satisfied when the conditional distribution of the simulated
520 data given $T$ equals that of the real data, which is a natural simulation target, though not
521 required for valid inference.

## D  How to Apply a Debiasing-based Approach

In addition to introducing our GMM-based estimator, we also study how debiasing-based methods, commonly referred to as prediction-powered inference (PPI) (Angelopoulos et al., 2023a) in the machine learning literature, can be adapted to our setting. Debiasing-based methods, which are a family of methods used in the literature for incorporating biased sources of information, have been well-studied in the context of predicted outcomes and, more recently, predicted covariates (i.e., proxy data). However, it is not immediately clear how to incorporate *fully* synthetic data and aggregate multiple sources of information (i.e., proxy data and synthetic data) in this setup. Perhaps the most general approach is given by RePPI (Ji et al., 2025), which predicts the optimal loss through fitting an arbitrary model that maps the proxy and synthetic loss to the real loss. In order to limit the number of parameters, we examine a natural instantiation of this, where the model is a convex combination.

**Proposition 2.** *The adapted, debiasing-based loss objective with multiple predicted covariates and outcomes is given by*

$$L^{PP}(\theta) := \frac{1}{N} \sum_{i=1}^{N} [(1-\alpha) \cdot l_\theta(\tilde{X}_i, \tilde{Y}_i) + \alpha \cdot l_\theta(\hat{X}_i, \hat{Y}_i)] \tag{6}$$

$$+ \frac{1}{n} \sum_{i=1}^{n} (l_\theta(X_i, Y_i) - [(1-\alpha) \cdot l_\theta(\tilde{X}_i, \tilde{Y}_i) \tag{7}$$

$$+ \alpha \cdot l_\theta(\hat{X}_i, \hat{Y}_i)]). \tag{8}$$

*where the estimate retains asymptotic normality conditions (see Appendix G for the proof and algorithm details).*

Importantly, note that the addition of this hyperparameter $\alpha$ adds increased complexity, and techniques such as cross-fitting must be used to select it in a statistically valid fashion. We refer to the estimator with $\alpha = 1$ as **PPI++Proxy**, as the synthetic terms vanish, yielding an estimator that combines real and proxy data. We refer to the estimator with tunable $\alpha \in [0, 1]$ as **PPI++Synth**, which combines real, proxy, and synthetic data. We note that our implementation builds on PPI++ (Angelopoulos et al., 2023b), retaining all additional benefits, such as power tuning, over the standard PPI estimator.

## E  Additional Experimental Details

### E.1  Prompt Texts

We present the full text prompts that were used to generate proxy covariates and labels (for the proxy data) and synthetic data. Note that the prompts used to extract covariates and labels from the synthetic text are identical to those used for the proxy data.

---

**Proxy Data Generation Prompts**

**Politeness (First Plural Pronouns) - Covariates:**
Does the following text contain first person plural pronouns (e.g., we, us, our, ourselves)? Output either yes or no.
Text: """
{content}
"""
**Answer:**

**Politeness (First Plural Pronouns) - Labels:**
Is the following text polite? Output either A or B. Output a letter only.
A) Polite
B) Impolite
Text: """
{content}
"""
**Answer:**

**Politeness (Hedging) - Covariates:**
Does the following text contain hedging devices—expressions that indicate uncertainty, caution, or a lack of full commitment to a claim (e.g., may, might, could, would, possibly, probably, perhaps, apparently, suggest, indicate, seem, appear, it is likely that, it seems that)? Respond with yes or no only.
Text: """
{content}
"""
**Answer:**

**Politeness (Hedging) - Labels:**
Is the following text polite? Output either A or B. Output a letter only.
A) Polite
B) Impolite
Text: """
{content}
"""
**Answer:**

**Stance Dataset - Covariates:**
Does the following text contain any affirmative device words? Output either yes or no.
Text: """
{content}
"""
**Answer:**

550

---

---

**Proxy Data Generation Prompts (continued)**

**Stance Dataset - Labels:**
A statement can agree, be neutral, or disagree with the statement: "Climate change/global warming is a serious concern". Classify the following statement into one of the three categories. Output either A, B, or C. Output a letter only.
A) Agree
B) Neutral
C) Disagree
Statement: """
{content}
"""
**Answer:**

**Congressional Bills Dataset - Covariates:**
You are a political scientist familiar with the U.S. Congress and the DW-NOMINATE scoring system, which places legislators and legislation on a left-right ideological spectrum ranging approximately from -1 (most liberal) to +1 (most conservative). Below is the text of a proposed bill. Based on the policy content, language, and framing of the bill, estimate the DW-NOMINATE score that best represents its ideological position. Output a single nonzero float between -1 and +1 representing the estimated DW-NOMINATE score of the bill.
Bill: """
{content}
"""
**Answer:**

**Congressional Bills Dataset - Labels:**
Does the following text relate to the economy? Output either true or false.
Text: """
{content}
"""
**Label:**

551

---

---

**Synthetic Data Generation Prompts**

**Politeness (First Plural Pronouns)**
Consider texts taken from user requests on Stack Exchange or Wikipedia. Each text is labeled as either polite or impolite, and either contains or does not contain first-person plural pronouns. Below is an example that {x}:
**Example:** """
{example}
"""

Now, generate a new example of a request that also {x}.

**Politeness (Hedging)**
Consider texts taken from user requests on Stack Exchange or Wikipedia. Each text can be labeled as either polite or impolite, and as either containing a hedging device or not containing one. Hedging devices are expressions that indicate uncertainty, caution, or a lack of full commitment to a claim (e.g., may, might, could, would, possibly, probably, perhaps, apparently, suggest, indicate, etc.). Below is an example that {x}:
**Example:** """
{example}
"""

Now, generate a new example of a request that also {x}.

552

---

**Synthetic Data Generation Prompts (continued)**

**Stance**
Consider news headlines that take a stance — agree, disagree, or neutral — on the statement: "Climate change/global warming is a serious concern."
Each headline also either contains or does not contain an affirmative device.
Below is an example of a headline.
**Example:** """
{example}
"""
Affirmative device: {x}
Now, generate a new news headline about global warming that also {x}.

**Congressional Bills Data**
You are a political language model trained to generate realistic examples of U.S. congressional bills. Each bill is labeled as either "related to the economy" or "not related to the economy", and is associated with a DW-NOMINATE score representing ideological position (ranging from $-1$ liberal to $+1$ conservative).
**Example:**
Bill Text: """
{example}
"""
DW-NOMINATE Score: {dw_nominate_score}
Now, generate a new example of a bill that also has a DW-NOMINATE score of {dw_nominate_score}. Output only the new bill text: """

553

---

## F   Additional Experimental Results

We present additional experimental results consisting of: (1) grid search of debiased-based approaches (Figure 3); (2) effective sample size analysis (Figure 4); (3) performance of a naive estimator that *only* uses synthetic data (Figure 5); and (4) cross-fitting results for our adapted debiasing approach (Figure 6).

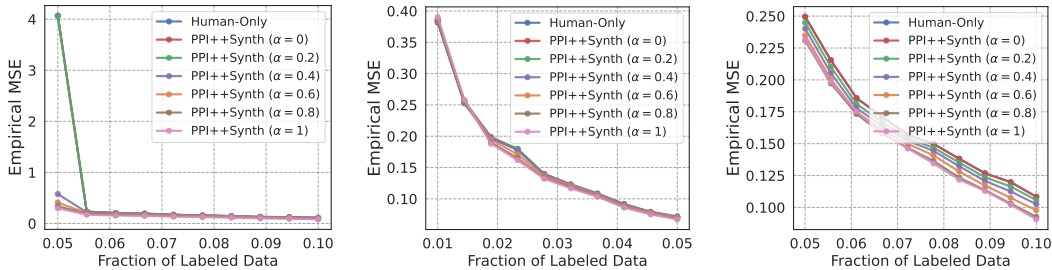

Figure 3: Grid search of the proposed debiasing-based approach (PPI++Synth) across different $\alpha$ values (on 1PP, Hedging, and Stance estimation tasks (from left to right)). We can observe that the optimal $\alpha$ value amongst the ones searched is defaulted to 1 in all cases, which is equivalent to collapsing to fully using the proxy data. Results are averaged over 200 trials.

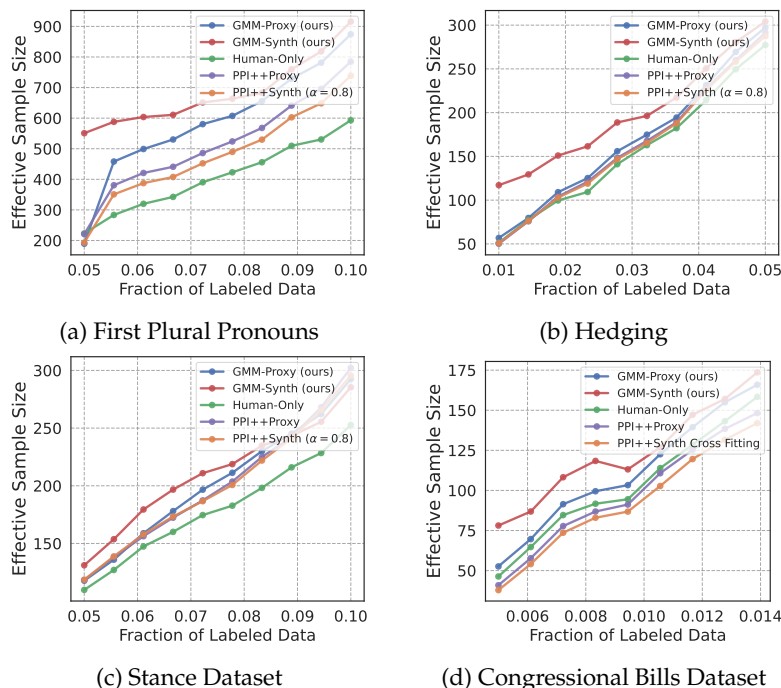

Figure 4: Effective sample size. We further evaluate our approach and baselines in terms of effective sample size, which quantifies the reduction in human annotations required to achieve estimates of equivalent accuracy.

In Figure 3, as an upper bound, we conduct a grid search over different possible $\alpha$ values *without* cross-fitting. Note, this is not a valid solution (and just an oracle comparison) as it requires hyperparameter tuning with access to the held-out data. In Figure 3, we empirically find that although this oracle incorporates proxy data effectively, introducing the synthetic data still does not yield further performance improvement; the optimal $\alpha$ is 1 in all cases, which is equivalent to only utilizing information from the proxy data terms (i.e., ignoring the synthetic data terms completely).

In Figure 4, we observe that our GMM-based approaches consistently improve performance in terms of effective sample size across all tasks. That is, our method reduces the number of human annotations needed to achieve *equally accurate* estimates. This is particularly valuable in label-scarce settings, highlighting its practical value for practitioners in low-resource, limited-labeled settings.

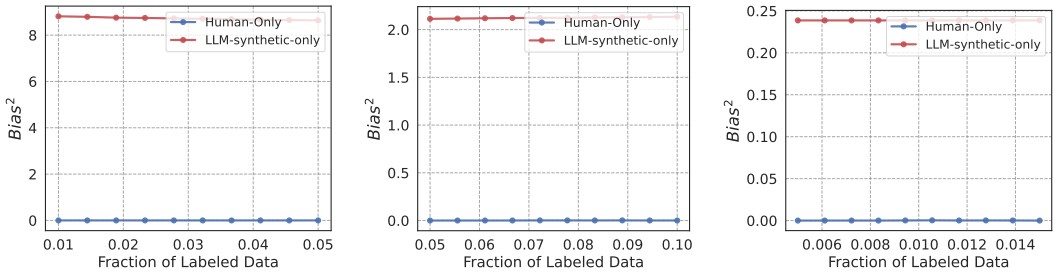

Figure 5: Performance of a naive estimator using synthetic data only (Politeness (Hedging), Stance, Congressional Bills (from left to right)). We clearly observe that naively using only synthetic data for the estimation task leads to largely biased estimates, as expected.

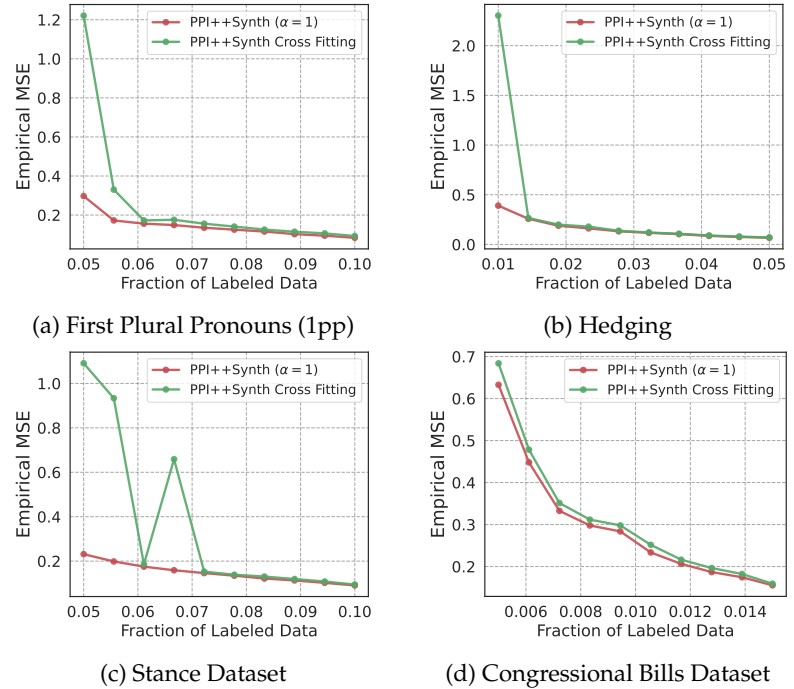

Figure 6: Cross-fitting results for PPI++Synth. We include $\alpha = 1$ as a reference point, which is equivalent to using only the proxy data.

## G  Adapted Debiasing-based Approaches: PPI++Proxy and PPI++Synth

We now present a discussion on our adapted debiasing-based approach from Proposition 2.

### G.1  Asymptotic Normality

First, it is relatively straightforward to show that this is an unbiased estimate of the true objective.

$$\mathbb{E}[L^{PP}(\theta)] = (1 - \alpha) \cdot \mathbb{E}[l_\theta(\tilde{X}, \tilde{Y})] + \alpha \cdot \mathbb{E}[l_\theta(\hat{X}, \hat{Y})]$$
$$+ \ \mathbb{E}[l_\theta(X, Y)] - \mathbb{E}[(1 - \alpha) \cdot l_\theta(\tilde{X}, \tilde{Y})] - \alpha \cdot \mathbb{E}[l_\theta(\hat{X}, \hat{Y})])]$$
$$= \mathbb{E}[\ell_\theta(X, Y)].$$

Note that this holds for any choice of the hyperparameter $\alpha$.

Under the same assumptions as in the PPI++ paper (Angelopoulos et al., 2023b) (e.g., that $\frac{n}{n+m} \to c$ for some constant $c$ and, in the case of generalized linear models, the Hessian is

---

**Algorithm 1** Cross-Fitting for PPI$^{++}$Synth

---

**Require:**
1: Labeled data $\mathcal{D} = \{(T_i, X_i, Y_i)\}_{i=1}^{n}$,
2: Proxy data $\widehat{\mathcal{D}} = \{(T_j, \widehat{X}_j, \widehat{Y}_j)\}_{j=1}^{n+m}$,
3: Synthetic data $\widetilde{\mathcal{D}} = \{(\widetilde{T}_j, \widetilde{X}_j, \widetilde{Y}_j)\}_{j=1}^{n+m}$,
4: K folds
**Ensure:** Debiased estimate $\hat{\theta}_{\text{CF}}$
5: Split $\mathcal{D}$ into folds $\{\mathcal{I}_1, \ldots, \mathcal{I}_K\}$
6:
7: **for** $k = 1, \ldots, K$ **do**
8:     define train-fold $\mathcal{I}_{\text{train}} = \bigcup_{r \neq k} \mathcal{I}_r$
9:     $\hat{\theta}_1^{-k} \leftarrow \arg\min_\theta L_{\text{PP}}^{-k}(\theta; 0)$                          ▷ (1) initial fit on train-fold
10:
11:    $\hat{\alpha}^{-k} \leftarrow \arg\min_{\alpha \in [0,1]} L_{\text{PP}}^{-k}(\hat{\theta}_1^{-k}; \alpha)$          ▷ (2) select mixture weight $\alpha$ on train-fold)
12:
13:    $\hat{\theta}^k \leftarrow \arg\min_\theta L_{\text{PP}}^{k}(\theta; \hat{\alpha}^{-k})$              ▷ (3) final fit on held-out fold with chosen $\alpha$)
14:
15: **end for**
16: **return** $\hat{\theta}_{\text{CF}} = \dfrac{1}{K} \sum_{k=1}^{K} \hat{\theta}^k$

---

non-singular, we perform their same approach to power tuning), we recover the asymptotic normality guarantees of the parameter estimate (as in Corollary 1 from Angelopoulos et al. (2023b)).

## G.2 Hyperparameter Selection via Cross-fitting

The added complexity from these modified debiasing-based approaches arises from the hyperparameter $\alpha$. We now discuss an approach for selecting $\alpha$ by performing cross-fitting. As previously mentioned, we can treat $\alpha$ as a simple version of RePPI (Ji et al., 2025) where we fit a convex combination of proxy and synthetic losses.

Namely, we partition our available data into two splits. We select $\alpha$ on one fold by minimizing:

$$\arg\min_{\alpha \in [0,1]} L^{PP}(\theta_1),$$

where $\theta_1$ is defined as the solution to the naive minimzation of $\mathbb{E}[\ell_\theta(X, Y)]$ on the same split. This essentially captures picking the $\alpha$ that best combines the proxy and synthetic losses to best mimic the behavior of the standard loss function.

We then take this optimal $\alpha$ and use it to produce a parameter estimate on the held-out fold. We aggregate these estimates as is standard in cross-fitting approaches. We outline this process in Algorithm 1.

