# OpenReview forum: "Drawing Reliable Conclusions with Synthetic Simulations from Large Language Models"
_colmweb.org/COLM/2025/Workshop/Social_Sim — Social Sim'25_

### Official Review · Reviewer_9gBT · 2025-07-15
**this submission proposes a new method to augmenting real data with synthetic data, that automatically weighs the synthetic data based on its usefulness**

**Rating:** 9
**Overall Assessment:** 4
**Confidence:** 4

**Review:**

Although I have not fully wrapped by head yet around the exact mechanism by which the proposed approach is able to automatically extract useful information from synthetic data and what it does with this information, it seems to me that the author of this submission are potentially onto something major. They effectively propose a way to automatically detect the relevance of synthetic data and extract just the right information from it. Intuitively, it seems to me that the method may also be compared conceptually to setting up a system of Seemingly Unrelated Regressions, where some regressions come from real data and others come from proxy and/or synthetic data paired with the real data, and that the gains offered by the proposed method are akin to the gains offered by capturing correlated error terms in such systems of equations. In that case, I wonder whether the benefit of the proposed method is primarily efficiency (i.e., smaller standard errors) or also accuracy.

In sum, there seems to be a novel and potentially impactful idea in this submission. I invite the authors to work on exposition to clearly convey the essence of the proposed approach and the exact mechanism by which it provides value. A couple of very simple "toy" examples may help.

**Comments Suggestions And Typos:**

None

**Paper Summary:**

This submission tackles the challenges of leveraging LLMs to improve the estimation of relevant quantities (e.g., the extent to which the use of pronouns predict politeness), in cases in which researchers have access to limited training data. The basic approach summarized in Figure 1 may not appear particularly novel at first glance: use the LLMs to generate labels for unlabeled texts and to generate new texts together with labels. I believe the innovation comes from (i) specifically generating synthetic examples that are paired with real examples (e.g., "here is an example of text with X and T, generate a new example that matches its style") and (ii) how the real data is combined with the synthetic data. In particular, the submission proposes a GMM approach that first estimates the relevant parameters "separately" for the real and synthetic examples, and then jointly re-estimates the two sets of parameters in a way that leverage the correlation structure between the residuals from the first step. This way, the synthetic data influences the relevant parameter estimates only when the residuals from the synthetic data are correlated with the residuals from the real data. The submission demonstrates using various simulations that this approach improves our ability to estimate relevant parameters.

**Relevance:**

5

**Summary Of Strengths:**

The originality of the submission is potentially great. And the potential usefulness is also very significant, given this is a general solution to an increasingly common problem.

**Summary Of Weaknesses:**

The submission may not be very easily accessible to readers who are not expert econometricians. The writing is already good, but the intuition behind the proposed method could be conveyed even more clearly.

---

### Official Review · Reviewer_zJvh · 2025-07-17

**Rating:** 9
**Overall Assessment:** 4
**Confidence:** 3

**Review:**

Overall, this paper is well-motivated, well-written, and could potentially address a quite important topic of the social sims community. There are some questions around experimental design and framing (see below). P.s. I am unable to assess the correctness of the method itself so my review will be focused on other aspect of the paper, and my confidence score reflects this.

**Comments Suggestions And Typos:**

None

**Paper Summary:**

This paper proposes a framework to reliably incorporate synthetic data + label into downstream statistical analysis by making use of the generalized method of moments.

**Relevance:**

4

**Summary Of Strengths:**

Novel angle and interesting result.

**Summary Of Weaknesses:**

1.	The term "social simulation" (used in the abstract and intro) may create a expectations mismatch for readers familiar with simulating human experiment type of simulation. The paper's method is more accurately described as something like synthetic data augmentation for statistical estimation.
2.	Missing some very relevant work such as https://papers.ssrn.com/sol3/papers.cfm?abstract_id=5133034
3.	The experiment is only done with gpt4o. What if the underlying model is stronger or weaker? How sensitive is the proposed framework towards that? This point needs to be empirically shown.

---

### Meta-Review · Program_Chairs · 2025-07-24

**Recommendation:** Accept

**Metareview:**

--